Morphological and molecular systematic review of Marphysa Quatrefages, 1865 (Annelida: Eunicidae) species from South Africa

Kara Jyothi 1 2
Molina-Acevedo Isabel C. imolina@ecosur.edu.mx 3 4
Zanol Joana 5
Simon Carol 1
Idris Izwandy 3
1 Department of Botany and Zoology, Stellenbosch University , Private Bag X1, Matieland , Stellenbosch , South Africa
2 Research and Exhibitions Department, Iziko Museums of South Africa , Cape Town , South Africa
3 South China Sea Repository and Reference Centre, Institute of Oceanography and Environment, Universiti Malaysia Terengganu , Kuala Nerus , Terengganu , Malaysia
4 Estructura y Función del Bentos, Depto. de Sistemática y Ecología Acuática., El Colegio de la Frontera Sur , Chetumal , Quintana Roo , México
5 Departamento de Invertebrados, Museu Nacional, Universidade Federal do Rio de Janeiro , Quinta da Boa Vista , São Cristovão , Brazil
Reimer James
Electronic publication date: 2020 Oct 21
Publication date: 2020
Volume: 8
Electronic Location ID: e10076
Received 2020 May 19; Accepted 2020 Sep 10
Copyright: ©2020 Kara et al.
Copyright year: 2020
Copyright holder: Kara et al.
License: This is an open access article distributed under the terms of the Creative Commons Attribution License, which permits unrestricted use, distribution, reproduction and adaptation in any medium and for any purpose provided that it is properly attributed. For attribution, the original author(s), title, publication source (PeerJ) and either DOI or URL of the article must be cited.
License URL: https://creativecommons.org/licenses/by/4.0/

Keywords: COI sequences, Wide distribution, Morphology, New species, Diversity

Funding: National Research Foundation 116654 CONACyT 514117/298079 Jyothi Kara was supported by a Postdoctoral Fellowship from the National Research Foundation (Grant Number: 116654) and Isabel C. Molina-Acevedo was supported by a scholarship from CONACyT (514117/298079). The funders had no role in study design, data collection and analysis, decision to publish, or preparation of the manuscript.

==============================
A vast polychaete fauna is hidden behind complexes of cryptic and pseudo-cryptic species, which has greatly hindered our understanding of species diversity in several regions worldwide. Among the eunicids, Marphysa sanguinea Montagu, 1813 is a typical example, recorded in three oceans and with various species considered its junior synonyms. In South Africa, specimens previously misidentified as M. sanguinea are now known as Marphysa elityeni Lewis & Karageorgopoulos, 2008. Of the six Marphysa Quatrefages, 1865a species recorded from the same region, three have their distributions restricted to South Africa while the others are considered to have worldwide distributions. Here, we evaluated the taxonomic status of the indigenous M. elityeni and investigated the presence of the widespread species Marphysa macintoshi Crossland, 1903 and Marphysa depressa Schmarda, 1861 in South Africa using morphological and molecular data. Our results reveal that M. elityeni is a junior synonym of Marphysa haemasoma, a species previously described from South Africa which is herein reinstated as a valid species. Both M. macintoshi and M. depressa are not present in South Africa and their status as being distributed worldwide deserves further investigation. Marphysa durbanensis Day, 1934 and the new species described here, M. sherlockae n. sp., had been misidentified as M. macintoshi and M. depressa respectively. Thus, the number of Marphysa species with distributions restricted to South Africa increased from three to five. This study reiterates the importance of implementing an integrated taxonomic framework to unravel local biodiversity.

Introduction

Studies implementing molecular and morphological tools in an integrated framework have found that a large portion of polychaete diversity has been hidden among complexes of cryptic and pseudo-cryptic species (Knowlton, 1993; Nygren, 2014; Hutchings & Kupriyanova, 2018). Thus, unravelling these species complexes can uncover patterns of distribution, regional biodiversity, and areas of endemism of previously overlooked polychaete species, which could have management and conservation implications (Bickford et al., 2007; Nygren, 2014).

Species belonging to Marphysa Quatrefages, 1865a and Quatrefages, 1865b, which serve as important bait species around the world (Izuka, 1912; Lewis & Karageorgopoulos, 2008; Idris, Hutchings & Arshad, 2014; Liu, Hutchings & Sun, 2017; Lavesque et al., 2017; Watson et al., 2017; Cole, Chick & Hutchings, 2018; Martin et al., 2020), are ideal candidates to investigate the incidence of complexes of pseudo-cryptic species. These complexes are frequently a consequence of very brief original species descriptions, as for Marphysa sanguinea (Montagu, 1813), type species of the genus (Hutchings & Karageorgopoulos, 2003). As a result of the brief species description, several morphologically similar species from far-flung places globally were considered junior synonyms of M. sanguinea (Hutchings & Karageorgopoulos, 2003; Molina-Acevedo & Carrera-Parra, 2015). As a consequence, its already broad distribution range was expanded, and it was reported to occur in Spain (Parapar, Besteiro & Urgorri, 1993), South Africa (Day, 1967), Australia (Day, 1967), Mexican Caribbean (Salazar-Vallejo & Carrera-Parra, 1998) and Japan (Miura, 1986) among others.

However, the detailed redescription of M. sanguinea and designation of the neotype (Hutchings & Karageorgopoulos, 2003) resulted in the reinstatement of at least three junior synonyms as valid species, including M. acicularum (Webster, 1884), M. nobilis (Treadwell, 1917), and M. viridis (Treadwell, 1917) (e.g.,  Molina-Acevedo & Carrera-Parra, 2015; Molina-Acevedo & Idris, 2020). Furthermore, several new species with restricted distributions were described (e.g.,  Hutchings & Karageorgopoulos, 2003; Glasby & Hutchings, 2010; Zanol, Da Silva & Hutchings, 2016; Zanol, Da Silva & Hutchings, 2017; Liu, Hutchings & Sun, 2017; Martin et al., 2020), some of which had been erroneously identified as M. sanguinea (e.g.,  Hutchings & Karageorgopoulos, 2003; Lewis & Karageorgopoulos, 2008; Lavesque et al., 2017; Wang, Zhang & Qiu, 2018). Detailed observations of specimens demonstrated the variability in diagnostic characters, like branchial distribution, parapodia shape, types of pectinate chaetae, coloration and shape of subacicular hooks, for Marphysa species that had previously been overlooked. The above-mentioned characters may apply to other species such as M. teretiuscula (Schmarda, 1861) and M. macintoshi Crossland, 1903, which also have suspiciously wide distribution ranges (Treadwell, 1906; Read & Fauchald, 2018).

Six valid species belonging to Marphysa are currently recognized as present in South Africa. Three have type localities in South Africa; Marphysa capensis (Schmarda, 1861), Marphysa posteriobranchia Day (1967), and Marphysa elityeni Lewis & Karageorgopoulos (2008). The latter is commonly known as the “wonder worm” by local fishermen, and is part of the global M. sanguinea species complex (Day, 1967; Lewis & Karageorgopoulos, 2008; Simon, Sato-Okoshi & Abe, 2019). The remaining three Marphysa species recorded in the region, namely M. corallina (Kinberg, 1865), M. depressa (Schmarda, 1861), and M. macintoshi Crossland, 1903 have type localities outside of South Africa and wide distributions (Day, 1967). Marphysa depressa’s type locality is in Auckland, New Zealand (Schmarda, 1861), and has since been recorded in Hong Kong (Wang, Zhang & Qiu, 2018) and South African estuaries from Saldanha Bay to Durban Bay (Day, 1953; Day, 1967). Marphysa macintoshi was described from Zanzibar (Crossland, 1903) and has since been recorded from several localities including Australia, South Africa, Caribbean Sea, Mozambique, Red Sea, Trinidad and Tobago and China (Read & Fauchald, 2018). In South Africa, this species is supposedly present from Cape St. Francis to Durban Bay (Day, 1967). Interestingly, M. durbanensis (Day, 1934) described from KwaZulu-Natal in South Africa, is considered a junior synonym of M. macintoshi (Day, 1967). Similarly, M. haemasoma Quatrefages, 1866 was described from Table Bay in South Africa and is currently considered a junior synonym of M. sanguinea. Thus, both species probably represent valid indigenous species that were incorrectly synonymized.

In this study, we investigated whether M. depressa and M. macintoshi occur in South Africa and examined the taxonomic validity of M. haemasoma. These were achieved by conducting thorough taxonomic revisions and, where possible, molecular comparisons. We also provide redescriptions of M. haemasoma, M. durbanensis, and a description of M. sherlockae n. sp., a species new to science from South Africa.

Material and Methods

Examined material

Fresh Marphysa depressa-like specimens were collected from rock crevices in the fringing intertidal zones from Strand (−34.116108, 18.821698) (n = 4) (Fig. 1). Fresh specimens of M. elityeni were collected from the fringing intertidal zone at low tide from burrows in gravely-sand type sediment under boulders in Kommetjie (n = 5) (−34.159709, 18.327851) (Fig. 1). Full collection data for both species can be found in the respective species accounts in the ‘Results’. Live specimens were brought back to the laboratory where they were anesthetized with 7% MgCl2 in distilled water, and photographed. Whole specimens from Strand were fixed in 96% ethanol. Posterior ends of the Kommetjie specimens were fixed in 96% ethanol, while the anterior ends were fixed in a 4% seawater-formalin solution. The collection of live material was approved by The Department of Agriculture, Forestry and Fisheries in South Africa under the permit number RES2019/49. Type and non-type material of M. depressa, M. macintoshi, M. durbanensis, M. haemasoma and M. elityeni deposited at the Natural History Museum, London (BMNH), Museum National d’Histoire Naturelle, Paris (MNHN), the Natural History Museum, Vienna, Austria (NHMW) and the Iziko South African Museum (SAM) were examined.

Figure 1 Sampling localities of M. depressa (Langebaan, Strand), M. macintoshi (Durban Bay), M. haemasoma (Table Bay) and M. elityeni (Buffels Bay and Kommetjie) from South Africa.

Triangles represent museum material examined and circles represent samples collected in 2017 (M. elityeni–Kommetjie) and 2019 (M. depressa–Strand).

Morphological examination

Species descriptions were produced based on the type material, but a variation section with all specimens reviewed was also included.

The general structures such as the prostomium, peristomium, anterior region of the body, maxillary apparatus, branchiae, parapodia, chaetae, and pygidium were included in the descriptions. A dorsal incision was made in the specimen to extract and describe the maxillary apparatus, after which it was returned to its original position. The maxillary formula (MF) and measurements were taken according to Molina-Acevedo & Carrera-Parra (2015) and Molina-Acevedo & Carrera-Parra (2017). Six parapodia (three from the anterior region, two from the median, and one from the posterior region) were dissected to describe the morphology of the cirri and lobes, and simple and compound chaetae.

The chaetigers where branchiae and subacicular hooks start were indicated depending on the side where they began (‘L’ for Left, ’R’ for Right) with the chaetiger number. In the region with the maximum number of branchial filaments, the long filaments are ≥4 times as long as dorsal cirri, whereas the short filaments are <4 times as long as dorsal cirri. The terminology used for the descriptions of the pectinate chaetae is according to the classification proposed by Molina-Acevedo & Carrera-Parra (2015), Molina-Acevedo & Carrera-Parra (2017) and Zanol, Da Silva & Hutchings (2016). Herein, thin and thick refers to the thickness of the pectinate shaft; wide and narrow refers to the width of the pectinate blade; and anodont and isodont refer to the relative length of external teeth in relation to each other and internal teeth, e.g., thin, wide isodont with long and slender teeth.

The length through chaetiger 10 (L10) and the width of chaetiger 10 excluding parapodia (W10) were measured in the specimens as standard measures when the specimens were collected incomplete. Likewise, the total length (TL) and variations of the total number of chaetigers (TChae) were recorded. All descriptions were illustrated with a series of photos taken with Canon EOS T6i. These were then stacked using Helicon Focus® 6 (Method A) software to improve the depth of field, and the final editing was performed in Adobe Photoshop® 2020.

To understand patterns of intraspecific variation, linear regression analyses were conducted to evaluate the possible relationships between size (length of specimens using L10 measurement) and morphological features such as the chaetigers where branchiae or the subacicular hooks begin and the number of branchial filaments. The degree of predictability of variation in morphological features following size variation is given by R2 (e.g., R2 = 0.63, p = 0.05, n = 34).

The electronic version of this article in Portable Document Format (PDF) will represent a published work according to the International Commission on Zoological Nomenclature (ICZN), and hence the new names contained in the electronic version are effectively published under that Code from the electronic edition alone. This published work and the nomenclatural acts it contains have been registered in ZooBank, the online registration system for the ICZN. The ZooBank LSIDs (Life Science Identifiers) can be resolved and the associated information viewed through any standard web browser by appending the LSID to the prefix http://zoobank.org/. The LSID for this publication is urn:lsid:zoobank.org:pub:C4C08B70-EC42-4AE1-9F9A-FDC717142D35. The online version of this work is archived and available from the following digital repositories: PeerJ, PubMed Central and CLOCKSS.

Molecular methods

DNA extraction, amplification and sequencing

DNA was extracted from tissue samples using the ZR Genomic DNA Tissue MiniPrep Kit according to the standard manufacturer’s protocol. The universal primer pair LCO1490 and HCO2198 (Folmer et al., 1994) was used to amplify a fragment of the mitochondrial gene cytochrome oxidase I (COI). PCR amplifications were carried out using 12.5 µl of OneTaq Quick-Load Master Mix (New England BioLabs), 9.5 µl of molecular biology grade water, 0.50 µl of forward and reverse primer (10 µM), 1 µl of 1% bovine serum albumin (BSA) and 1 µl of template DNA to make up a total reaction volume of 25 µl. Thermal cycling conditions were as follows for M. elityeni and M. sherlockae n. sp.: initial denaturation at 95 °C for 3 minutes, followed by 35 cycles of 94 °C for 20 seconds, 45 °C for 30 seconds and 72 °C for 1 minute, followed by a final extension time at 72 °C for 5 minutes. Amplicons were Sanger sequenced at the Central Analytical Facility at Stellenbosch University using just the forward primer (LCO1490). Quality control was performed on sequences to check for any sequencing errors using BioEdit (v7.2.6) (Hall, 1999).

Phylogenetic and species delimitation methods

The COI sequences were edited, trimmed, and aligned with ClustalW (Thompson, Higgins & Gibson, 1994) using multiple alignment methods in BioEdit (v7.2.6). Several species belonging to the Marphysa genus were included in the analysis for comparison together with seven other species from different genera within the Eunicidae and one species from Onuphidae as they were used as outgroups to root the tree (see Table 1). DnaSP v5 (Librado & Rozas, 2009) was used to generate a nexus file for subsequent analysis. PAUP (Swofford, 2003) and MrModelTest v2.3 (Nylander, 2004) were used to calculate the best fit model of evolution for the data set using the Aikaike Information Criterion (AIC). Bayesian inference (BI) was used to reconstruct phylogenetic relationships using the best fit model SYM+G in MrBayes 3.1.2 (Ronquist et al., 2012). The trees were calculated using 4 Markov Chains of 5 million generations sampled simultaneously with every 1000th tree sampled. A 50% majority-rule consensus tree with posterior probability support was constructed by discarding the first 25% of trees as burn-in. Tracer v1.5 (Rambaut & Drummond, 2009) was used to investigate the convergence of runs by analysing the average standard deviation of split frequencies (≤ 0.01). The mixing quality of all parameters was verified by analyzing the plot of likelihood versus the sampled trees and the effective sample sizes (ESS >200), of which both criteria were satisfied. FigTree v1.4.4 (Rambaut, 2012) was used to visualize trees. A Maximum Likelihood tree was computed in MEGA X (Kumar et al., 2018) and was run for 500 bootstrap replicates using the best-fit model of evolution, GTR, that was calculated in the same program.

Table 1 COI sequences of Marphysa taxa used in the phylogenetic analysis, with GenBank accession numbers, museum voucher numbers, type locality of species, location of specimens from which sequences were obtained and references to the respective studies.

Species	Genbank accession number	Type locality (TL)	Collection locality	Reference	
Marphysa haemasoma	MN067877	Cape of Good Hope, South Africa	Kommetjie, South Africa	Simon et al. unpublished data.	
Marphysa sherlockaen. sp.	MT840349 –MT840351	Durban, South Africa	Strand, South Africa	This study	
Marphysa aegypti	MF196971, MF196969, MF196970, MF196968	Suez Canal, Egypt	Suez Canal, Egypt	Elgetany et al. (2018)	
Marphysa chirigota	MN816444, MN816442, MN816443	Cádiz Bay, SW Iberian Península	Cádiz Bay, SW Iberian Península	Martin et al. (2020)	
Marphysa bifurcata	KX172177, KX172178	Sheltered North Reef at Poin Peron, Western Australia	Australia	Zanol, Da Silva & Hutchings (2016)	
Marphysa brevitentaculata	GQ497548	Scarborough, Tobago, Trinidad and Tobago	Mexico	Zanol et al. (2010)	
Marphysa californica	GQ497552	San Diego County, California	California	Zanol et al. (2010)	
Marphysa corallina	KT823271, KT823300, KT823306, KT823343, KT823371, KT823389, KT823410	Hawaii	Multiple localities in KwaZulu-Natal and Eastern Cape, South Africa	Kara et al. unpublished	
Marphysa fauchaldi	KX172165	off Elizabeth River, Darwin region, Australia	Australia	Zanol, Da Silva & Hutchings (2016)	
Marphysa gaditana	MN816441, KR916870, AY040708, KR916871, KR916872, KR91687, KP254503, KP254537, KP254643, KP254743, KP254802	Cádiz Bay, SW Iberian Península	Cádiz Bay, SW Iberian Península Portugal, France, Virginia (USA)	Martin et al. (2020), Lobo et al. (2016), Siddal et al. (2001) and Leray & Knowlton (2015)	
Marphysa honkongensa	MH598526	Tolo Harbour, Hong Kong	China	Wang, Zhang & Qiu (2018)	
Marphysa iloiloensis	MN133418, MN106279, MN106280, MN106281	Tigbauan, Iloilo Province	Philippines	Glasby et al. (2019)	
Marphysa kristiani	KX172141, KX172142, KX172143, KX172144, KX172145, KX172146, KX172147, KX172148, KX172149, KX172150, KX172151, KX172155, KX172152, KX172153, KX172154, KX172156, KX172157, KX172158, KX172159, KX172160, KX172161, KX172162, KX172163	Stingray Bay, New South Wales	Australia	Zanol, Da Silva & Hutchings (2016)	
Marphysa mossambica	JX559751, KX172164	Mozambique	Philippines, Australia	Zanol et al. (2010) and Zanol, Da Silva & Hutchings (2016)	
Marphysa mullawa	KX172166, KX172167, KX172168, KX172169, KX172170, KX172171, KX172172, KX172173, KX172174, KX172175, KX172176	Moreton Bay, Fisherman’s Island, Queensland	Australia	Zanol, Da Silva & Hutchings (2016)	
Marphysa pseudosessiloa	KY605405, KY605406	Careel Bay, New South Wales	Australia	Zanol, Da Silva & Hutchings (2017)	
Marphysa regalis	GQ497562	Bermuda	Brazil	Zanol, Da Silva & Hutchings (2016)	
Marphysa victori	MG384996, MG384999, MG384997, MG384998	Arcachon Bay	France	Lavesque et al. (2017)	
Marphysa viridis	GQ497553	Boca Grande Key, Florida	Brazil	Zanol et al. (2010)	
Marphysa sanguinea	GQ497547, MK541904, MK950851, MK950852, MK950853, MK967470, MN106282, MN106283, MN106284	Polperro, Cornwall	Cornwall (UK), France	Zanol et al. (2010), Lavesque et al. (2019) and Glasby et al. (2019)	
Marphysa tripectinata	MN106271, MN10622, MN1062723, MN106274, MN106275, MN106276, MN106277, MN106278	Beihai, China	China	Liu, Hutchings & Sun (2017)	
Marphysasp.	KP255196, KP254890, KP254644, KP254223, NC023124, KF733802	–	Florida (USA), China	Leray & Knowlton (2015) and Li et al. (2016)	
Paucibranchia bellii	KT307661	Chausey Island, France	Spain	Aylagas et al. (2016)	
Paucibranchia disjuncta	GQ497549	Los Angeles County, California	California, USA	Zanol et al. (2010)	
Paucibranchiasp.	JX559753		Phillipines	Zanol et al. (2014)	
Palola viridis	GQ497556	Samoa	Micronesia	Zanol et al. (2010)	
Eunice cf. violaceomaculata	GQ497542	–	Belize	Zanol et al. (2010)	
Leodice rubra	GQ497528	–	Brazil	Zanol et al. (2010)	
Hyalinoeciasp.	GQ497524	–	Massachusetts, USA	Zanol et al. (2010)	

A Newick formatted phylogenetic tree generated using FigTree v1.4.4 from the previous analysis was used as input for the Bayesian implementation of the Poisson tree process (bPTP) (Zhang et al., 2013) model for species delimitation using the online webserver https://species.h-its.org/. The tree was rooted and run for 500,000 MCMC generations, with thinning set to 100 and burn-in and seed set to 0.1 and 123, respectively. The convergence of MCMC chains was visually checked on the maximum likelihood plot generated by the online server.

MEGA X was used to calculate the interspecific genetic distances between species using the Kimura 2-parameter (K2P) model with complete deletion of gaps.

Results

Thorough morphological comparisons indicate that M. macintoshi and M. depressa do not occur in South Africa. Instead, M. durbanensis (type locality: South Africa), which was previously synonymized with M. macintoshi (type locality: Tanzania/Zanzibar) (Day, 1967) has been found to differ from the latter species with regards to the shape of the prostomium, anterior postchaetal lobes, pectinate chaetae, and the shape and distribution of branchiae throughout the length of the body. As a result, we herewith consider M. durbanensis as a valid species.

Moreover, specimens initially identified as M. depressa (type locality: New Zealand) in South Africa were a misidentified and instead represent a new species to science, whereby named M. sherlockae n. sp.. Morphological comparisons reveal that M. sherlockae n. sp. differs from M. depressa in the shape and distribution of compound chaetae, the shape of postchaetal lobes, and the maximum number of branchial filaments. COI sequences of M. depressa were not available from its type locality and could not be compared with sequences of M. sherlockae n. sp. Nonetheless, M. sherlockae n. sp. forms an independent phylogenetic clade with high posterior probability and bootstrap support (Fig. 2) and genetically differs from other Marphysa species included in the phylogenetic analysis by 18–25%, confirming that it is a separate species. Additionally, results from the bPTP analysis supported M. sherlockae n. sp. as a single independent species (BS>0.95) (S1, supplementary information). M. sherlockae n. sp. is phylogenetically closest to Marphysa californica Moore, 1909, and Marphysa brevitentaculata, but the clade is poorly supported. Nonetheless, all three species genetically differ from each other by 18–20%.

Figure 2 Bayesian phylogenetic tree based on the mitochondrial cytochrome c oxidase subunit 1 alignment of Marphysa spp.

Bayesian posterior probabilities and maximum likelihood bootstrap values >95% are represented by an * at each node with the former on the left side of forward slash and latter on the right side of the forward slash. The hyphens, –, represent nodes that had maximum likelihood bootstrap values <90%. Purple clade—the reinstated M. haemasoma. Blue clade—newly described M. sherlockae n. sp.

Marphysa haemasoma is a valid species. The examination of type materials allowed us to confirm that M. haemasoma differs from M. sanguinea in the shape of the postchaetal lobe in anterior chaetigers and subacicular hooks, the maximum number of branchial filaments and in the distribution of the swollen base of ventral cirri. Furthermore, types of M. elityeni only differ from those of M. haemasoma in size-related features, such as the length of prostomial appendices, and where branchiae and ventral cirri with a swollen base start. For these reasons, and in view of the principle of priority (ICZN, 1999, Arts. 23), we consider Marphysa haemasoma a senior synonym of M. elityeni. Furthermore, M. haemasoma forms a well-supported phylogenetic clade independent of the M. sanguinea clade (Fig. 2). The species are genetically different from each other by 20%, with results from the bPTP analysis (S1 supplementary information), confirming their separation as independent species (BS>0.95). Thus, these species are not synonymous.

Systematics

Order EUNICIDA Dales, 1962	
Family EUNICIDAE Berthold, 1827	
Genus MarphysaQuatrefages, 1865a	

Marphysa durbanensisDay, 1934

Figures 3, 4A and 5

Marphysa durbanensis Day, 1934:51–53, text-Fig. 10.

Marphysa macintoshi— Day, 1967:378 (non Crossland, 1903; Day, 1974:59; Branch et al., 2016:68–69, Pl. 26, Fig. 26.6.

Material examined. Type material: lectotype designated here BMNH 1934.1.19.166, Durban, South Africa, 1933, coll. JH. Day. One paralectotype BMNH 2020.39 designated here, same information as lectotype.

Comparative material examined. Marphysa macintoshi, syntypes, three specimens, BMNH 1924.3.1.22-3, slide BMNH.1924.3.1.22A, Zanzibar, Africa, 1901-1902, by digging in sand between intertidal on both east and west coasts of Zanzibar (syntype 1 incomplete specimen with 262 chaetigers, L10: 8.1 mm, W10: 2.7 mm; syntype 2 incomplete specimen with 106 chaetigers, L10: 5.3 mm, W10: 3 mm; syntype 3 incomplete specimen with 160 chaetigers, L10: 7.8 mm, W10: 3 mm).

Description. Lectotype complete, ventrally dissected from peristomium until chaetiger 9, with 380 chaetigers, L10 = 14 mm, W10 = 3.6 mm, TL = 305 mm. Last 48 chaetigers regenerating. Anterior region of body with convex dorsum and flat ventrum; body depressed from chaetiger 7, widest at chaetiger 24, tapering after chaetiger 37.

Prostomium bilobed, 1.7 mm long, 2.5 mm wide; lobes anteriorly rounded; median sulcus shallow dorsally (Fig. 3A), deep ventrally (Fig. 3B). Prostomial appendages in a semicircle, median antenna isolated by a gap. Palps reaching middle of first peristomial ring; lateral antennae reaching middle of second peristomial ring; median antenna broken, in paralectotype reaching middle of first chaetiger. Palpophores and ceratophores ring-shaped, short, thick; palpostyles and ceratostyles tapering, slender. Eyes not observed.

Figure 3 Marphysa durbanensisDay, 1934.

(A) Anterior end, dorsal view; (B) Anterior end, ventral view; (C) Anterior view, lateral view; (D) Maxillary apparatus, dorsal view; (E) Left MI-II-III-IV-V, lateral view; (F) Mandible; (G) Parapodium 3; (H) Parapodium 8; (I) Parapodium 12; (J) Parapodium 69; (K) Parapodium 217; (L) Thin narrow isodont pectinate with long and slender teeth, chaetiger 3; (M) Thin wide isodont with short and slender teeth, chaetiger 69; (N) Thick wide isodont pectinate with short and thick teeth, chaetiger 140; (O) Thick wide anodont with short and slender teeth, chaetiger 140; (P) Compound spinigers, chaetiger 3; (Q) Subacicular hook, chaetiger 278. A–C, G–P from lectotype BMNH 1934.1.19.166; D–F, Q from paralectotype BMNH 2020.39. All chaetigers in anterior view; LMI-II: Ligament between MI and MII; LMII-III: Ligament between MII and MIII. Scale bars: A–C, 3.5 mm; D–E, 0.9 mm; F, 0.8 mm; G–K, 0.2 mm; L–O, Q 30 µm; P, 0.1 mm.

Figure 4 Distribution of branchial filaments throughout the body in (A) Marphysa durbanensisDay, 1934 (L10: 14 mm, 380 chaetigers); (B) Marphysa haemasomaQuatrefages, 1865a (L10: 123 mm, 322 chaetigers); (C) Marphysa sherlockaen. sp. (L10: 6.6 mm, 208 chaetigers).

Figure 5 Marphysa durbanensisDay, 1934, lectotype BMNH 1934.1.19.166.

(A) Thin narrow isodont pectinate with long and slender teeth, chaetiger 3; (B) Thin wide isodont with short and slender teeth, chaetiger 69; (C) Thick wide isodont pectinate with short and thick teeth, chaetiger 140; (D) Thick wide anodont with short and slender teeth, chaetiger 140. Scale bars: A–D, 30 µm.

Peristomium (2.7 mm long, 3 mm wide) longer and wider than prostomium, first ring twice as long as second ring; separation between rings distinct on all sides (Figs. 3A–3C). Ventral anterior edge of peristomium longer than dorsal, remaining features ventrally distorted by the dissection (Figs. 3B–3C).

Maxillary apparatus with MF= 1+1, 5+6, 6+0, 4+8, 1+1 (Fig. 3D). MI 3.1 times longer than maxillary carriers. MI forceps-like, MI 4.6 times longer than closing system (Figs. 3D–3E); ligament between MI and MII sclerotized. MII wider than rest of maxillae, with triangular teeth; MII 3.6 times longer than cavity opening oval (Figs. 3D–3E); ligament present between MII–MIII and right MII–MIV slightly sclerotized (Fig. 3E). MIII with triangular teeth; with rectangular attachment lamella, situated in the centre of ventral edge of maxilla, slightly sclerotized (Figs. 3D–3E). Left MIV with two left-most teeth larger; attachment lamella semicircle, slender, better developed in central portion, situated 1/2 along anterior edge of maxilla. Right MIV with teeth of equal size; attachment lamella semicircle, slender, better developed in central portion, situated 2/3 along anterior edge of maxilla, sclerotized (Figs. 3D–3E). MV square, with a short triangular tooth. Mandibles dark; missing calcareous cutting plates; sclerotized cutting plates brown, with 20 growth rings (Fig. 3F).

Branchiae pectinate with up to 11 long filaments at around 64–80% of the body, present from chaetigers 28L–29R to 370 (Figs. 3J–3K). First pair and last 10 with one filament; reach the maximum 10 or 11 filaments in chaetigers 241L–307L (Fig. 4A). Branchial filaments longer than dorsal cirri except in first five and last seven branchiae.

First two parapodia smallest; best developed in chaetigers 6–26, following ones becoming gradually smaller. Notopodial cirri conical in anterior-median chaetigers, digitiform in posterior ones; longer than ventral cirri in anterior chaetigers, of similar length in posterior ones; best developed in chaetigers 3–30, following ones gradually smaller (Figs. 3G–3K). Prechaetal lobes short, as transverse fold in all chaetigers (Figs. 3G–3K). Chaetal lobes rounded in all chaetigers, shorter than postchaetal lobes in anterior region, longer than the other lobes in median-posterior region; with aciculae emerging dorsal to midline (Figs. 3G–3K). Postchaetal lobes well developed in first 40 chaetigers; digitiform in first five chaetigers, rounded from chaetiger 6; progressively smaller from chaetiger 22; from chaetiger 41 inconspicuous (Figs. 3G–3K). Ventral cirri bluntly conical in first five chaetigers; in chaetigers 6 to 355 with a short oval base and digitiform tip; conical from chaetiger 356, gradually reducing in size (Figs. 3G–3K).

Aciculae blunt, reddish along most of their length, amber on the distal tip (Figs. 3G–3K). First eight chaetigers with three aciculae; in chaetigers 9–18 with four aciculae; in chaetigers 19–44 with three or four aciculae; in chaetigers 45–124 with two aciculae; from chaetiger 125 with only one acicula.

Limbate chaetae of two lengths in same chaetiger, dorsalmost longer; reduced in number around chaetiger 30. Five types of pectinate chaetae, anterior chaetigers: thin, narrow isodont with long and slender teeth, 3–4 pectinate, with up to 14–15 teeth (Figs. 3L and 3A); median and posterior chaetigers: thin, wide isodont with short and slender teeth, 4–5 pectinate, with up to 23–24 teeth (Figs. 3M and 3B); thick, wide isodont with short and thick teeth, 1–2 pectinate, with up 19 teeth (Figs. 3N and 3C); and thick wide anodont with short and slender teeth, 1–2 pectinate, with 12 teeth (Figs. 3O and 3D); posterior chaetigers: thick, wide anodont with long and thick teeth, 1–2 pectinate, with up to 17 teeth. Compound spinigers present in all chaetigers, in anterior-median chaetigers with blades of two lengths, shorter ones more abundant (Fig. 3P). Subacicular hooks unidentate, amber, present from chaetiger 46, one or two per chaetiger, with continuous distribution (Fig. 3Q).

Pygidium with dorsal pair of anal cirri as long as last eight chaetigers; ventral pair short, as long as last two chaetigers.

Variations. Material examined L10= 12–14 mm, W10= 3.6–4 mm, TChae= 322–380. Palps reaching middle of first or second peristomial ring; lateral antennae reaching middle of second peristomial ring or first chaetiger; median antenna reaching first chaetiger. The maxillary variations are MII 5–6+6–8, MIII 6, MIV 3–4+6–8. The proportion of maxillary apparatus varies as follows: MI are 3.1–3.2 times longer than maxillary carriers; MI are 4.6–5.3 times longer than closing system; MII are 3.5–3.6 times longer than length of cavity opening. Branchiae from chaetigers 28–32 to 10–13 chaetigers before pygidium. Maximum number of branchial filaments varied from 11 to 12. Postchaetal lobe well developed in the first 40 chaetigers. Ventral cirri with a swollen base from chaetigers 4–5 to 25 chaetigers before pygidium. Start of subacicular hooks in chaetigers 46–47.

Habitat. Day (1934) does not provide information about the specific substrate, although he did clarify that the collection was between the tidemarks in Durban Bay and Umkomaas.

Distribution. Day (1934) recorded this species from Durban Bay and Umkomaas in KwaZulu-Natal, South Africa.

Remarks. The original description of Marphysa durbanensis provides a variation of the two specimens collected that matches with the specimens deposited in the BMNH. Day (1934) described almost colorless eyes, but they were not observed in this study. Possibly the color has faded due to the long-term preservation of the specimens. The best-preserved specimen is herein selected as a lectotype to fix the species definition (ICZN, 1999, Arts. 74.1, 74.7.3), whereas the other is considered a paralectotype (ICZN, 1999, Art. 74F).

Day (1934) considered M. durbanensis different from morphologically similar species such as M. simplex Crossland, 1903 (= M. teretiuscula), and M. acicularum when he described the species. However, in his monograph of the polychaetes from South Africa, the author considered M. durbanensis a junior synonym of M. macintoshi without making any reference to this nomenclatural action (Day, 1967, page 378). Herein, apparent differences were found between the species. Marphysa durbanensis (L10: 14 mm) has a bilobed prostomium, the branchiae are pectinate and start from chaetigers 28–32, the postchaetal lobe is digitiform in first four chaetigers, and there are five types of pectinate chaetae; while in M. macintoshi (L10: 4.5 mm) the prostomium is unilobed with a shallow median sulcus at the anterior edge, the branchiae are palmate with a short button-shaped branchial stem and start from chaetiger 32–47, the postchaetal lobe is conical in the first four chaetigers, and there are only three types of pectinate chaetae. Due to these morphological differences, M. durbanensis is considered a valid species.

Marphysa durbanensis resembles M. haemasoma (see below) by the presence of compound spinigers distributed in all chaetigers; however, M. durbanensis has more teeth in MII (5–6+6–8), digitiform postchaetal lobes in first four chaetigers, five types of pectinate chaetae, and the subacicular hook with a continuous distribution even in bigger specimens. However, M. haemasoma has fewer teeth in MII (4+4). The postchaetal lobe is ovoid in the first four chaetigers. There are only four types of pectinate chaetae, and the subacicular hook has a discontinuous distribution in small specimens.

Marphysa durbanensis resembles M. victori Lavesque et al., 2017, M. hongkongensa Wang, Zhang & Qiu, 2018, M. leidii Quatrefages, 1866, M. parishii Baird, 1869 and M. teretiuscula by the presence of five types of pectinate chaetae; however, M. durbanensis has a digitiform postchaetal lobe in the first four chaetigers, and the subacicular hook is amber, while M. teretiuscula has an ovoid postchaetal lobe in the first four chaetigers, and the subacicular hook is reddish basally and translucent in the distal region. Also, M. leidii has a conical postchaetal lobe in the first chaetigers. Otherwise, M. durbanensis has long branchial filaments, and the branchiae are pectinate; while for M. hongkongensa, the branchial filaments are short, and the branchiae are pectinate and palmate with a short button-shaped branchial stem in some regions of the body. On the other hand, in M. durbanensis (L10: 14 mm), the eyes are present, and the branchiae start in chaetigers 28–32; while M. victori (L10: 6.3–7.9 mm) lacks eyes, and the branchiae start in chaetiger 36. Finally, M. durbanensis has up to 11–12 branchial filaments while M. leidii (L10: 10.7–17 mm) and M. parishii (L10: 17.2 mm) only have 4 to 6 filaments.

Marphysa haemasomaQuatrefages, 1866

Figures 4B, 6–7

Figure 6 Marphysa haemasomaQuatrefages, 1866.

(A) Anterior end, dorsal view; (B) Anterior end, ventral view; (C) Anterior view, lateral view; (D) Maxillary apparatus, dorsal view; (E) Left MI-II-III-IV-V, lateral view; (F) Mandible; (G) Parapodium 3; (H) Parapodium 12; (I) Parapodium 30; (J) Parapodium 154; (K) Parapodium 307; (L) Thin narrow isodont with long and slender teeth, chaetiger 3; (M) Thick wide isodont with short and slender teeth, chaetiger 251; (N) Thick wide anodont with short and slender teeth, chaetiger 307; (O) Thick wide anodont with long and thick teeth, chaetiger 251; (P) Compound spinigers, chaetiger 3; (Q) Subacicular hook, chaetiger 209. A–B, D–E, G–L, N, P from holotype M. haemasoma MNHN type 613; F, M, O, Q from paratype Marphysa elityeni BMNH 2007.69. All chaetigers in anterior view; al-MIII: attachment lamella MIII; al-MIV: attachment lamella MIV; LMI-II: Ligament between MI and MII; LMII-III: Ligament between MII and MIII. Scale bars: A–B, 3.1 mm; C, 3.8 mm; D–E, 1.2 mm; F, 1.7 mm; G–K, 0.2 mm; L–O, Q, 30 µm; P, 0.1 mm.

Figure 7 Marphysa haemasomaQuatrefages, 1866. Type and additional material from Marphysa elityeniLewis & Karageorgopoulos, 2008.

(A) Anterior end, dorsal view; (B) Left MI-II-III-IV-V, lateral view; (C) Right MI-II-IV-V, lateral view; (D) Parapodium 3; (E) Parapodium 13; (F) Parapodium 208; (G) Parapodium 3; (H) Parapodium 12; (I) Thick wide anodont with short and slender teeth, chaetiger 209; (J) Thick wide isodont with short and slender teeth, chaetiger 209; (K) Subacicular hook, chaetiger 209. A–F, from paratype Marphysa elityeni BMNH 2007.69; G–H from holotype M. haemasoma MNHN type 613; I–K, from topotype M. elityeni BMNH 237. Chaetigers D-E, G-H in posterior view, chaetiger F in anterior view; al-MIII: attachment lamella MIII; al-MIV: attachment lamella MIV; LMI-II: Ligament between MI and MII; LMII-III: Ligament between MII and MIII; LMII-IV: Ligament between MII and MIV; PL: Postchaetal lobe. Scale bars: A, 4.6 mm; B–C, 1.8 mm; D–E, 0.4 mm; G–H, 0.2 mm; I–J, 30 µm.

Marphysa haemasoma Quatrefages, 1866:334–335; Grube, 1870:299.

Marphysa sanguinea— von Marenzeller, 1888:11, Fauvel, 1902:61; Day, 1967:378 (non Montagu, 1813); Day, 1974:59.

Marphysa sanguinea haemasoma Willey, 1904:263, Pl.13, Fig.15

Marphysa elityeni Lewis & Karageorgopoulos, 2008:279–281, Figs. 1–2, Tables 1, 2, 3; Branch et al., 2016:68–69, Pl. 26, Fig. 26.5.

Material examined. Type material: holotype Marphysa haemasoma MNHN type 613, Cape of Good Hope, South Africa. Additional material: Five incomplete specimens SAM-A090272, SAM-A090273, SAM-A090274, SAM-A090275, SAM-A090317, Kommetjie, South Africa from sand burrows under boulders at fringing intertidal zone, coll. A.N. du Toit, 10 Mar 2017, 18°19′40.7″E 34°09′33.0″S.

Comparative material examined. Holotype Marphysa elityeni SAM-A21478, Cape of Good Hope, South Africa. Eight paratypes of Marphysa elityeni BMNH 2007.69, SAM-A21479, SAM-A21480, SAM-A21481, Buffels Bay in the Cape of Good Hope, South Africa, 15 Sep 2004, 18°29′27″E 34°21′6″S. Neotype Marphysa sanguinea BMNH 1867.1.7.24, Polperro, Cornwall, in mud and gravel at low water mark, coll. Laughrin, Redet. P. Hutchings (2 specimens from this lot), Desig. P. Hutchings (Neotype complete specimen with 286 chaetigers, L10: 16.7 mm, W10: 10 mm; topotype complete specimen with 239 chaetigers, L10: 20.4 mm, W10: 7.2 mm).

Description. Holotype complete, gravid female, with 322 chaetigers, L10 = 12.3 mm, W10 = 7 mm TL = 309 mm. Anterior region of the body with convex dorsum and flat venter; body depressed from chaetiger 5, widest at chaetiger 25, tapering after chaetiger 41.

Prostomium bilobed, 2.8 mm long, 4 mm wide; lobes anteriorly rounded; median sulcus dorsally shallow (Fig. 6A), ventrally deep (Fig. 6B). Prostomial appendages in a semicircle, median antenna isolated by a gap. Palps reaching first chaetiger; lateral and median antennae reaching second chaetiger. Palpophores and ceratophores ring-shaped, short, thick; palpostyles and ceratostyles tapering, slender. Eyes colorless, as a scar between palps and lateral antennae.

Peristomium (2.8 mm long, 6.3 mm wide) wider than prostomium; first ring three times as long as second ring, separation between rings distinct only dorsally and ventrally (Figs. 6A–6C). Ventral region of the first ring with a slight central depression in anterior edge (Fig. 6B).

Maxillary apparatus with MF = 1 + 1, 4 + 4, 5 + 0, 3 + 7, 1 + 1 (Fig. 6D). MI 3 times longer than maxillary carriers. MI forceps-like, MI 4 times longer than closing system (Figs. 6D–6E); ligament between MI and MII, sclerotized. MII with triangular teeth, right anterior teeth broken; MII 3.6 times longer than cavity opening (Figs. 6D–6E); ligament present between MII–MIII and right MII–MIV slightly sclerotized (Fig. 6E). MIII with triangular teeth; with rectangular attachment lamella, situated only in the centre of right edge of maxilla, slightly sclerotized (Figs. 6D–6E). Left MIV with all teeth of similar size; attachment lamella semicircle, wide, better developed in right portion, situated 2/3 of anterior edge of maxilla. Right MIV with lateral larger teeth; attachment lamella semicircle, wide, better developed in central portion, situated 2/3 of anterior edge of maxilla, sclerotized (Figs. 6D–6E). MV square, with a short triangular tooth. Mandibles dark; with calcareous cutting plates present and sclerotized cutting plates brown, with nine growth rings (Fig. 6F).

Branchiae pectinate with up to six long filaments for around 20–54% of the body, present from chaetigers 26L–27R to 308L–311R (Figs. 6I–6J). First two and last 13 pairs with one filament; with six filaments in chaetigers 79L to 173L (Fig. 4B). Branchial filaments longer than dorsal cirri except in first two and last branchiae.

First two parapodia smallest; best developed in chaetigers 7–40, following ones gradually becoming smaller. Notopodial cirri conical in all chaetigers; of similar length as ventral cirri in anterior and posterior chaetigers, shorter than ventral cirri in median chaetigers; best developed in chaetigers 4–37, following ones gradually smaller (Figs. 6G–6K). Prechaetal lobes short, as transverse folds in all chaetigers (Figs. 6G–6K). Chaetal lobes in first 37 chaetigers rounded, shorter than postchaetal lobe in anterior region, with aciculae emerging dorsal to midline; from chaetiger 38 triangular, longer than other lobes in median-posterior chaetigers (Figs. 6G–6K). Postchaetal lobes well developed in first 60 chaetigers; ovoid in first six chaetigers, rounded in chaetigers 7–9, auricular from chaetiger 10, progressively smaller from chaetiger 35; from chaetiger 61 inconspicuous (Figs. 6G–6K). Ventral cirri digitiform in first three chaetigers; in chaetiger four to last chaetiger with a short oval base and digitiform tip (Figs. 6G–6K).

Aciculae blunt, reddish along most of their length, amber on distal tip (Figs. 6G–6K). First 10 chaetigers with three aciculae; in chaetigers 11–77 with three or four; in chaetigers 78–161 with three; in chaetigers 162–322 with two or three.

Limbate chaetae of two lengths in same chaetiger, dorsalmost longer, reduced in number around chaetiger 24. Four types of pectinate chaetae; in anterior chaetigers: thin, narrow isodont with long and slender teeth, with 2–3 pectinate, with up to 17 teeth (Fig. 6L); median-posterior chaetigers: thick, wide isodont with short and slender teeth, with 6–7 pectinate, with up to 17 teeth (Fig. 6M); posterior chaetigers: thick, wide anodont with short and slender teeth, with 6–7 pectinate, with up to 13–14 teeth (Fig. 6N), and thick, wide anodont with long and thick teeth, with 1–2 pectinate, with up to 10 teeth (Fig. 6O). Compound spinigers present in all chaetigers, with blades of two sizes in the same chaetiger (Fig. 6P), shorter slightly more abundant than longer blade. Subacicular hooks absent; in paratype of M. elityeni (L10= 9.3 mm) subacicular hook bidentate, translucent, present only in regenerating chaetigers, one per chaetiger; with triangular teeth, distal tooth smaller than proximal, directed upward; proximal tooth triangular, directed laterally (Fig. 6Q).

Pygidium with dorsal pair of anal cirri broken; ventral pair as long as last chaetiger.

Variations. Material examined L10= 9.3–20.1 mm, W10= 6.2–14.5 mm, TChae= 194–486. Palps reaching second peristomial ring or first chaetiger; lateral antennae reaching first or second chaetiger; median antenna reaching first or middle of second chaetiger. The maxillary variations are MII 4+4, MIII 3–5, MIV 3–4+6–7. The proportion of maxillary apparatus varies as follows: MI are 2.6–3 times longer than maxillary carriers; MI are 4.1–4.6 times longer than closing system; MII are 4–4.3 times longer than cavity opening. Branchiae from chaetigers 26–37 to 10 chaetigers before pygidium. Maximum number of branchial filaments varied from six to 10. Postchaetal lobe well developed in first 57–60 chaetigers. Ventral cirri with a swollen base from chaetigers 3–6 to last chaetigers.

DNA barcode. Type locality: Kommetjie, Western Cape, South Africa (MB-A090272) (GenBank accession number: MN067877) (Simon et al. unpublished data). 577 bp fragment isolated with universal mitochondrial cytochrome oxidase subunit 1 gene, primer pair: LCO1490, HCO2198 (Folmer et al., 1994).

Habitat. Very common in the boulder fields at the lower intertidal zones of sheltered bays, and in rock pools. Worms can be found under rocks in sand burrows up to 1 m deep.

Distribution. Table Bay to Buffels Bay, Cape Point, Western Cape South Africa (Quatrefages, 1866; Lewis & Karageorgopoulos, 2008). Branch et al. (2016) recorded this species to occur from Namibia in southwest Africa to East London in South Africa. Simon et al. (unpublished data) recorded this species from Melkbosstrand to Knysna in the Western Cape and therefore falls within the currently accepted distribution range of this species according to Branch et al. (2016). However, the records from Namibia have not been verified and may also represent an overlooked indigenous species of that region and therefore should be revised.

Remarks. Specimens of M. haemasoma were previously redescribed by Grube (1870) and then identified as M. sanguinea after von Marenzeller, 1888 synonymized M. haemasoma with M. sanguinea due to similarities in morphology and habitat observed in the specimens from the Cape of Good Hope. Later, Lewis & Karageorgopoulos (2008) realized that specimens from this region had been misidentified as M. sanguinea, which led to the description of Marphysa elityeni Lewis & Karageorgopoulos, 2008. However, Lewis & Karageorgopoulos (2008) overlooked M. haemasoma.

After the comparison between the type material of M. haemasoma and M. elityeni we found stable similarities in the shape of the prostomium (Figs. 6A and 7A), the proportions of maxillary apparatus, the number of teeth per maxilla and the shape of the maxillary apparatus (Figs. 6D–6E, 7B–7C), the form of the branchiae in median-posterior chaetigers (Figs. 6J and 6F), the shape of the dorsal cirri, ventral cirri, and postchaetal lobe in anterior chaetigers (Figs. 6G–6H, 7D–7E, 7G–7H), as well as the presence of the same type of pectinate chaetae (Figs. 6M–6N, 7I–7J) and compound chaetae, and the form and coloration of subacicular hook (Figs. 6Q and 6K). Some differences were related to the size dependence of characters, like the beginning of the branchiae, the number of filaments, and the development of the postchaetal lobe (M. elityeni material L10: 9.3–18.5 mm, branchiae from chaetiger 27–37, number of filaments from 6–10, ending of the postchaetal lobe from chaetiger 33–82; M. haemasoma material L10: 12.3 mm, branchiae from chaetiger 26, number of filaments reached 6, ending of the postchaetal lobe in chaetiger 60).

Marphysa haemasoma (L10: 9.3–18.5 mm) is considered a different species from M. sanguinea (L10:11.5–20.4) because the former has up to 10 branchial filaments, and ovoid postchaetal lobes in anterior chaetigers; whereas the latter has 9–18 branchial filaments, and digitiform postchaetal lobes in anterior chaetigers. Moreover, in M. haemasoma the swollen base of the ventral cirri continues until the last chaetigers, and the subacicular hook is translucent; while in M. sanguinea the swollen base of the ventral cirri ends between 8–18 chaetigers before the pygidium, and the subacicular hook is reddish basally and translucent distally.

Marphysa haemasoma resembles M. aegypti Elgetany, El-Ghobashy, Ghoneim & Struck, 2018, M. fauchaldi Glasby & Hutchings, 2010, M. gravelyi Southern, 1921, M. nobilis Treadwell, 1917, M. teretiuscula Schmarda, 1861 and M. tripectinata Liu, Hutchings & Sun, 2017 by the presence of the ovoid postchaetal lobes; however, M. haemasoma has subacicular hooks that are completely translucent, while M. nobilis, M. teretiuscula, and M. tripectinata have subacicular hooks that are reddish at the base and translucent in the distal region. Furthermore, M. haemasoma has four types of pectinate chaetae, while M. fauchaldi and M. gravelyi have only three types. Additionally, when present in M. haemasoma, subacicular hooks (in regenerating chaetigers) are bidentate, while M. aegypti bears unidentate subacicular hooks (Martin et al., 2020). Moreover, M. haemasoma has fewer teeth in MII and MIII (4+4, 4–5+0), while M. gravelyi has more teeth in the same plates (MI 8+7, MII 8+0). Finally, M. haemasoma has long branchial filaments, while in M. fauchaldi, the branchial filaments are short.

Type material of M. elityeni was collected from Buffels Bay, Cape Peninsula (Lewis & Karageorgopoulos, 2008), which is ∼58.4 km away from Table Bay where type material of M. haemasoma was collected (Fig. 1). Additionally, Kommetjie, where the fresh materials examined and sequenced in this study were collected, is near both Buffels Bay (∼29.4 km) and Table Bay (∼43 km). Thus, all these collections fall within the type region of the original material collected from Table Bay (Fig. 1).

Marphysa sherlockae n. sp.

urn:lsid:zoobank.org:act:2D2AC893-C074-46CC-B731-F0D632C66836

Figures 4C, 8, 9 and 10

Figure 8 Marphysa sherlockae n. sp. Holotype BMNH 2020.40.

(A) Anterior end, dorsal view; (B) Anterior end, ventral view; (C) Anterior end, ventral view; (D) Maxillary apparatus, dorsal view; (E) Left MI-II-III-IV-V, lateral view; (F) Mandible; (G) Parapodium 3; (H) Parapodium 6; (I) Parapodium 14; (J) Parapodium 114; (K) Parapodium 185 (L) Thin narrow isodont with long and slender teeth, chaetiger 3; (M) Thick wide isodont with long and thick teeth, chaetiger 185; (N) Compound spinigers, chaetiger 3; (O) Compound falcigers, chaetiger 3; (P) Subacicular hook, chaetiger 49. All chaetigers in anterior view; al-MIII: attachment lamella MIII; al-MIV: attachment lamella MIV; LMI-II: Ligament between MI and MII; LMII-III: Ligament between MII and MIII. Scale bars: A–C, 1.7 mm; D–E, 0.6 mm; F, 0.4 mm; G–K, 0.1 mm; N–P, 30 µm.

Figure 9 Marphysa sherlockae n. sp.

(A) Thin narrow isodont pectinate chaetae with long and slender teeth, anterior chaetiger; (B) Thin narrow isodont pectinate chaetae with long and slender teeth, anterior chaetiger; (C) thin narrow isodont pectinate chaetae with long and slender teeth, chaetiger 32; (D) Thick wide isodont pectinate chaetae wide with long and thick teeth, posterior chaetiger; (E) Thick wide isodont pectinate chaetae with long and thick teeth, posterior chaetiger. A, B, C from SAMC- A20578; D, E SAMC-A089089 Scale bars: A–E, 0.05 mm.

Figure 10 Length-dependent variation of some morphological features in Marphyssa sherlockae n. sp.

Orange point: Chaetiger where subacicular hook start (p = 0.35, n = 11). Blue points: First chaetiger with branchia (p = 0.26, n = 11); Green points: Maximum number of branchial filaments (p = 0.00, n = 11).

Marphysa depressa— Day, 1953:434, text-Fig. 5 n, p; 1967:395–396, Fig. 17.5 n–t (non Schmarda, 1861); Day, 1974:59; Branch, Charles & King, 2016:68–69, Pl. 26, Fig. 26.8.

Material examined. Type material: holotype BMNH 2020.40, Langebaan Lagoon, South Africa, coll. J.H. Day. Paratype, one specimen BMNH 2020.41. Paratype, two specimens (SAMC-A089089 and SAMC-A089090), Strand, False Bay, South Africa, 34°06′57.9″S, 18°49′18.1″E, coll. J. Kara, 20 March 2019, det. J. Kara. Additional material: two specimens BMNH 1963.1.84, same data as holotype. One incomplete specimen SAMC-A20578, Langebaan lagoon, South Africa, coll. UCT ecological survey, 24 April 1949, det. J.H. Day. One complete specimen SAMC-A60425, Langebaan Lagoon, South Africa, coll. UCT ecological survey, 24 April 1949, det. D. Clarke. Two complete specimens, (SAMC- A089091 and SAMC- A089092), Strand, False Bay, South Africa, 34°06′57.9″S, 18°49′18.1″E, coll. J. Kara, 20 March 2019, det. J. Kara.

Comparative material examined. Syntypes, two specimens, Marphysa depressa NHM 1044, New Zealand, Port of Auckland, coll. Schmarda (syntype 1 complete specimen with 328 chaetigers, L10: 9.5, W10: 4 mm; syntype 2 complete specimen with 132 chaetigers, L10: 9.5 mm, W10: 4.8 mm).

Description. Holotype complete, with 208 chaetigers, L10= 6.6 mm, W10 = 1.7 mm, TL = 67 mm. Anterior region of body with convex dorsum and flat venter, body depressed from chaetiger 6, widest at chaetiger 38, tapering after chaetiger 112.

Prostomium bilobed, 1 mm long, 1.1 mm wide; lobes frontally oval; with median sulcus dorsally shallow (Fig. 8A), ventrally sulcus deep (Fig. 8B). Prostomial appendages in a semicircle, median antenna isolated by a gap. Palps reaching first chaetiger; lateral antennae reaching second chaetiger; median antenna reaching middle of second chaetiger. Palpophores and ceratophores ring-shaped, short, thick; palpostyles and ceratostyles tapering, slender. Eyes as a brown line, between palps and lateral antennae.

Peristomium (1.1 mm long, 3.2 mm wide) wider than prostomium, first ring twice as long as second, separation between rings distinct on all sides (Figs. 8A–8C). Ventral region of the first ring with a slight central depression in anterior edge (Fig. 8B).

Maxillary apparatus with MF= 1+1, 3+5, 5+0, 4+8, 1+1 (Fig. 6D). MI 2.3 times longer than maxillary carriers. MI forceps-like, MI 4.3 times longer than the closing system; ligament between MI and MII, slightly sclerotized (Figs. 8D–8E). MII with recurved teeth; MII five times longer than cavity opening oval (Figs. 8D–8E); ligament present between MII and MIII and right MIV slightly sclerotized (Fig. 8E). MIII with blunt teeth; with rectangular attachment lamella, situated in the anterior of right edge of maxilla, slightly sclerotized (Figs. 8D–8E). Left MIV with left-most tooth larger; attachment lamella semicircle, wide, better developed in right portion, situated along anterior edge of maxilla (Figs. 8D–8E). Right MIV with right-most tooth larger; attachment lamella semicircle, wide, better developed in central portion, situated along anterior edge of maxilla (Figs. 8D–8E). MV square, with a short-rounded tooth. Mandibles dark; missing calcareous cutting plates, sclerotized cutting plates brown, with 10 growth rings (Fig. 8F).

Branchiae palmate with a short button-shaped branchial stem, with up to two long filaments, present from chaetigers 28R–37L to 195L–196R (Figs. 8J–8K). One filament in chaetigers 28L and 31L–45L; without filament in chaetigers 29L–30L; two filaments in chaetigers 46L–170L; one filament in chaetigers171L–196L (Fig. 4C). Branchial filaments longer than dorsal cirri.

First two parapodia smallest; best developed in chaetigers 6–42, following ones becoming gradually smaller. Notopodial cirri conical in all chaetigers; longer than ventral cirri in anterior chaetigers, shorter in median chaetigers, of similar size in posterior ones; best developed in chaetigers 3–41, following ones gradually decreasing in size (Figs. 8G–8K). Prechaetal lobes short. Chaetal lobes in first 29 chaetigers rounded, shorter than postchaetal lobe, with aciculae emerging dorsal to midline; from chaetiger 30 triangular, longer than other lobes (Figs. 8G–8K). Postchaetal lobes slightly developed in first 24 chaetigers; triangular first five chaetigers, following ones auricular, progressively smaller from chaetiger eight; from chaetiger 25 inconspicuous (Figs. 8G–8K). Ventral cirri conical in first six chaetigers; from chaetigers 7 to 138 with a short oval base and digitiform tip; conical from chaetiger 139, gradually smaller (Figs. 8G–8K).

Aciculae blunt, reddish from base to most of its length, translucent on the distal tip (Figs. 8G–8K). First five chaetigers with two aciculae; in chaetiger 6–10 with three aciculae; in chaetigers 11–73 with two aciculae; from chaetiger 74 with only one acicula.

Limbate chaetae of two lengths in same chaetiger, dorsal-most longer, reduced in number around chaetiger 13. Two types of pectinate chaetae; in anterior chaetigers: thin, narrow isodont with long and slender teeth, 1–2 per parapodium and up to 10–11 teeth (Figs. 8L, 9A–9C); in median-posterior chaetigers, thick, wide isodont with long and thick teeth, 4–5 per parapodium and up to 14 teeth (Figs. 8M, 9D–9E); anodont pectinate not observed. Compound spiniger chaetae present in all chaetigers, with blades of similar size in the same chaetiger (Fig. 8N), longer blades in median-posterior chaetigers. Compound falciger chaetae in anterior-median chaetigers, more abundant than compound spiniger in first 26 chaetigers; in anterior region blades of similar length (56 µm, Fig. 8O), with triangular teeth, both of similar size, proximal tooth directed laterally, distal directed upward; in median chaetigers with blades shorter (38.5 µm) with teeth of similar shape. Subacicular hooks bidentate, reddish from base to most of its length, with translucent tip, starting from chaetigers 41R–42L, one per chaetiger, with continuous distribution; with blunt teeth, distal tooth smaller than proximal, both teeth directed upward (Fig. 8P); some chaetigers with subacicular hook unidentate with hoods.

Pygidium with dorsal pair of anal cirri as long as last seven chaetigers; ventral pair short, as long as the last chaetiger.

Variations. Material examined varied in the following features: L10 = 3–6.6 mm, W10 = 1.3–2.1 mm. Palps reaching second peristomial ring or first chaetiger; lateral antennae reaching middle of first or second chaetiger; median antenna reaching third or fourth chaetiger. Maxillary formula varies as follows: MII 3–4+4–5, MIII 5–6, MIV 3–4+7–8. The proportion of maxillary apparatus varies as follows: MI are 2.4–2.7 times longer than maxillary carriers; MI are 4.3–5 times longer than closing system; MII are 3–3.3 times longer than cavity opening. Branchiae from chaetigers 25–34. The maximum number of branchial filaments is two. Postchaetal lobe well developed in first 17–91 chaetigers. Ventral cirri with a swollen base from chaetigers 3–7 to 70 chaetigers before pygidium. Falcigers present up to last chaetiger (L10 = 3–6 mm) or median region (L10 = 6.1–66 mm). Start of subacicular hooks in chaetigers 28–43.

Regression analyses indicated that there are no correlations between the start of the branchiae (R2 = 0.0702, p = 0.26, n = 11, Fig. 10), the maximum number of branchial filaments (R2 = 0.000, p = 0.00 n = 11, Fig. 10) or the start of the subacicular hooks (R2 = 0.1307, p = 0.35, n = 11, Fig. 10) with the length to chaetiger 10 for this species. The chaetiger where the branchiae start does not follow a pattern regarding their growth but starts to emerge from chaetiger 20 to 30 (Fig. 8, blue points). This same situation is repeated with emergence of subacicular hooks, starts between chaetiger 30 and 40 (Fig. 10, orange points). However, the number of filaments (two filaments) seems to be fixed regardless of the size of the organism, a contrasting pattern with other Marphysa species in which the number of filaments appears to increase with the length of the specimen.

On the other hand, M. sherlockae n. sp. has similar characteristics to other species of Marphysa where the presence of compound chaetae is size-dependent (Aiyar, 1931; Pillai, 1958; Salazar-Vallejo & Carrera-Parra, 1998; Molina-Acevedo & Carrera-Parra, 2017; Molina-Acevedo, 2018). Marphysa sherlockae n. sp. specimens with L10 ≤ 6 mm possess compound falcigers to the last chaetiger. In this group of individuals, the number of falcigers per chaetiger decreased from median to posterior region, which was more noticeable in specimens with L10 close to 6 mm. Additionally, specimens with L10 >6 mm do not have falcigers in the posterior region. This condition indicates that in the largest specimens of M. sherlockae n. sp. falcigers will be lost, and only compound spinigers will be observed, as demonstrated in M. gravelyi Southern, 1921, M. borradailei Pillai, 1958 and M. brevitentaculata Treadwell, 1921.

Etymology: The species is named after Emma Sherlock, in recognition of her valuable work on the polychaete collections of BHNM.

DNA barcode: Type region: Strand, False Bay, Western Cape, South Africa (Museum number: SAMC-A089090) (GenBank accession number: MT840249). 577 bp fragment isolated with universal mitochondrial cytochrome oxidase subunit 1 gene, primer pair: LCO1490, HCO2198 (Folmer et al., 1994).

Habitat. Fringing rocky zones at low tide in sheltered bays. Worms can be found in rock crevices.

Type locality. Langebaan Lagoon, South Africa.

Distribution. Day (1953), Day (1967) and Branch et al. (2016) recorded this species to occur in rocky coasts and estuaries from Saldanha Bay in the Western Cape to Durban in KwaZulu-Natal, South Africa.

Remarks. Day (1953) studied the material collected by himself and other members of the Zoology Department at the University of Cape Town during ecological surveys of the rocky coasts and estuaries in South Africa. The author identified some specimens as Marphysa depressa collected from localities such as East London, Bushman’s Estuary, Still Bay, Cape Agulhas, and Langebaan Lagoon due to the presence of compound spinigers and falcigers in the same chaetiger which is similar to the New Zealand species. As a result, this was the first record of the species in South Africa. Additionally, Day compared his material with a specimen collected from New Zealand by Ehlers (1904), most likely to confirm his identification. However, thorough taxonomic revisions revealed marked differences between the material from South African and New Zealand and led us to conclude that the South African specimens belong to a new species named herein as Marphysa sherlockae n. sp.

Marphysa sherlockae n. sp. differs from M. depressa in the chaetal distribution. For example, the former has compound spinigers in all chaetigers, and compound falcigers restricted to the median and posterior chaetigers; whereas in M. depressa, the compound falciger is present in all chaetigers, but the spinigers are only present in the anterior region. Also, M. sherlockae n. sp. has a triangular postchaetal lobe, while M. depressa has a digitiform postchaetal lobe. Furthermore, M. sherlockae n. sp. (L10: 5.7–6.6 mm) has only two branchial filaments, while M. depressa (L10: 9.5 mm) has up to four filaments.

Marphysa sherlockae n. sp. resembles M. durbanensis and M. haemasoma by having compound spinigers. However, M. sherlockae n. sp. (L10: 5.7–6.6 mm) has two branchial filaments, triangular postchaetal lobe in anterior chaetigers, and ventral cirri with a swollen base ending 70 chaetigers before pygidium; whereas M. durbanensis (holotype, L10: 14 mm) has 11–12 branchial filaments, digitiform postchaetal lobes, and ventral cirri with a swollen base ending 25 chaetigers before pygidium. Further, M. haemasoma (L10: 9.3–18.5 mm) has 6–10 branchial filaments, ovoid postchaetal lobe, and ventral cirri with a swollen base until the last chaetiger.

Marphysa sherlockae n. sp. resembles M. angelensis Fauchald, 1970, M. brevitentaculata Treadwell, 1921, M. digitibranchia Hoagland, 1920, M. emiliae Molina-Acevedo & Carrera-Parra, 2017, M. formosa Steiner & Amaral, 2000, M. mangeri Augener, 1918, M. orensanzi Carrera-Parra & Salazar-Vallejo, 1998 and M. sebastiana Steiner & Amaral, 2000 by having compound falcigers and spinigers present; however, M. brevitentaculata, M. digitibranchia, and M. mangeri have limbate capillaries in the subacicular position from the middle to the posterior region of the body, while in M. sherlockae n. sp. these simple chaetae are absent. Furthermore, M. angelensis and M. emiliae have a digitiform postchaetal lobe in first four chaetigers, while in M. sherlockae n. sp. the postchaetal lobe is triangular at the same first chaetigers. Also, in M. emiliae (L10: 3.5–5.4 mm) branchiae begin in chaetigers 8–12; while in M. sherlockae n. sp. (L10: 3–6.6 mm) branchiae begin from 25–34. On the other hand, M. formosa has pectinate branchiae, while M. sherlockae n. sp. have palmate branchiae with a short button-shaped branchial stem. Furthermore, M. formosa (TL: 55 mm), M. orensanzi (TL: 12 mm), and M. sebastiana (LT: 120 mm) have up to 4–6 branchial filaments while M. sherlockae n. sp. (TL: 67 mm) only has two filaments. Finally, M. sebastiana and M. angelensis have short branchial filaments, while the filaments in M. sherlockae n. sp. are long.

Discussion

This study revealed that M. macintoshi and M. depressa recorded for the region actually represent (1) an incorrectly synonymized species, i.e., M. durbanensis that was reinstated herein, and (2) a new indigenous species that was previously overlooked and herein described, i.e., M. sherlockae n. sp., respectively. We also confirm the notion addressed by Lewis & Karageorgopoulos (2008), that M. sanguinea is not present along the South African coast. However, the local species should be named M. haemasoma Quatrefages, 1866 and not M. elityeni Lewis & Karageorgopoulos, 2008, since the latter is a junior synonym of the former.

Marphysa depressa and M. macintoshi were first recorded along the South African coast by Day (1953) and Day (1967) with summary descriptions and general illustrations. The recurrent identification of M. macintoshi and M. depressa along the South African coast (e.g., Branch, Charles & King, 2016) reflects the overlooking of detailed characteristics and the use of traditional and conspicuous diagnostic features considered enough to define Marphysa species, such as the color and shape of the subacicular hook, distribution of compound chaetae throughout the body, the shape and distribution of branchiae, and the number of branchial filaments (Quatrefages, 1866; Grube, 1878; McIntosh, 1910; Hartman, 1944; Fauchald, 1970, among others). The sole use of distinctive conspicuous features in the identification may lead to spurious records of cosmopolitanism in species (Hutchings & Kupriyanova, 2018), and also to the proliferation of misleading species records and synonymization.

The detailed study of the traditional conspicuous features, the discovery of new unique characters as well as the examination of type specimens, as carried out here, has improved the morphological delimitation of Marphysa species, and the understanding of the diversity within the genus (e.g., Glasby & Hutchings, 2010; Molina-Acevedo & Carrera-Parra, 2015; Molina-Acevedo & Carrera-Parra, 2017). Therefore, recent studies on Marphysa have focused on detecting unique characters or in the re-assessment of those forgotten features, such as the shapes of dorsal cirri, postchaetal lobes, and pectinate chaetae, and the first appearance of the ventral cirrus with a swollen base. For instance, Miura (1986) and Molina-Acevedo & Carrera-Parra (2015) have shown that the distribution of the number of filaments and the region where the maximum number is reached can be informative in species delimitation. Here, the distribution of branchial filaments is different in each analyzed species (Fig. 4). Thus, whenever possible, it should be incorporated in future descriptions of Marphysa species. The main challenge of using “new” features in taxonomic investigations is the lack of this information in older descriptions preventing comparison. Thus, the examination of type material deposited in museums or examining newly collected material from the type locality in cases where no types were deposited previously is an essential step towards improving the taxonomy and recognition of new or inappropriate synonyms as in the case of M. haemasoma.

Molecular data provide an additional source of information that improves our knowledge on species boundaries and aids in recognition of intraspecific variation (e.g., Lewis & Karageorgopoulos, 2008; Zanol, Da Silva & Hutchings, 2016; Zanol, Da Silva & Hutchings, 2017; Lavesque et al., 2017; Elgetany et al., 2018; Lavesque et al., 2019; Glasby et al., 2019; Abe et al., 2019; Martin et al., 2020). The phylogenetic tree revealed two distinct South African monophyletic clades, belonging to the new species M. sherlockae n. sp., and the other to M. haemasoma. The molecular analyses reinforced the re-establishment of M. haemasoma as a valid species by confirming its distinction from M. sanguinea, which concurs with previous findings from the region (Lewis & Karageorgopoulos, 2008). Furthermore, for the first time, this study provided COI sequences of M. haemasoma, from South Africa.

A total of nine Marphysa species have been newly proposed or redescribed under an integrative taxonomic framework since 2003 (Zanol, Da Silva & Hutchings, 2016; Zanol, Da Silva & Hutchings, 2017; Lavesque et al., 2017; Elgetany et al., 2018; Lavesque et al., 2019; Glasby et al., 2019; Abe et al., 2019; Martin et al., 2020; present study), thus, increasing the number of publicly available sequences of Marphysa species globally. This framework, in turn, provides a starting point from which other studies can address more complex hypotheses, such as resolving the phylogenetic placements of species within the genus.

This study has confirmed that the indigenous diversity of Marphysa in South Africa was indeed previously underestimated and thus increases the number of described indigenous species from three to five (Day, 1967; Lewis & Karageorgopoulos, 2008) and reduces the number of putative cosmopolitan species to one (i.e., Marphysa corallina). Similarly, studies by Lewis & Karageorgopoulos (2008), Clarke et al. (2010), Kara, Macdonald & Simon (2018) and Simon, Sato-Okoshi & Abe (2019) provide additional evidence that many cosmopolitan species reported in the Day (1967) polychaete monograph for this region are actually incorrect assignments. Undoubtedly, the polychaete monograph by Day (1967) is an invaluable resource for polychaete descriptions and distributions. However, it is widely used by researchers from many disciplines, including those working outside of the region (Hutchings & Kupriyanova, 2018). Thus, biologists locally and internationally should take cognizance of this fact and use the monograph with caution, especially concerning species considered “cosmopolitan”.

Using information from Day (1967) and Awad, Griffiths & Turpie (2002) determined that only 20% of polychaete species in South Africa are endemic to the region. Thus, if only half the remaining 80% prove to be misidentifications of indigenous species, our understanding of diversity, biogeography, and endemism of polychaete worms in South Africa has been severely underestimated, and priority conservation areas may need to be reviewed. Furthermore, the resolution of taxonomically confusing species, such as those belonging to Marphysa, and development of realistic diversity estimates will be improved if voucher specimens are deposited in museums for taxonomic and molecular investigations.

Conclusion

Marphysa in South Africa is represented by six species, namely, M. capensis, M. corallina, M. durbanensis, M. haemasoma, M. posteriobranchia, and M. sherlockae n. sp. Although the number of species is similar to previous identifications, the resurrection of M. haemasoma, synonymization of M. elityeni with M. haemasoma, reinstatement of M. durbanensis from M. macintoshi and redescription of M. sherlockae n. sp. from M. depressa has changed the composition of endemic and cosmopolitan species. As such, gaining a better understanding of our true local biodiversity may help us to understand the extent of biodiversity loss in the face of climate change and make better decisions regarding the designation of marine protected areas.

Supplemental Information

Data S1 Species delimitation results: mPTP

Results based on Bayesian Inference reconstruction

Click here for additional data file.

Data S2 Regression analysis data

Characters dependent on the size of Marphysa sherlockae n. sp. (Figure 6)

Click here for additional data file.

Data S3 Sequence data from M. sherlockae and M. haemasoma and M

Click here for additional data file.

Data S4 Branchial filaments distribution

Number of filaments per chaetiger in M. durbanensis, M. haemasoma and M. sherlockae n. sp. (Figure 7).

Click here for additional data file.

Data S5 Interspecific genetic distances between species

The analysis of interspecific genetic distances between species study in the phylogenetic analysis.

Click here for additional data file.

We sincerely thank Emma Sherlock (BMNH), Tarik Meziane (MNHN), Helmut Sattmann (NHMW), and Able Bosman (SAM) for making available some of the materials that made this study possible. To Dr. Luis F. Carrera-Parra and Dr. Sergio I. Salazar-Vallejo for their advice and conversations about the morphology of the Eunicidae family. To Biol. Humberto Bahena-Basave (ECOSUR, Mexico) for advice on digital photography and editing. We also thank editors and reviewers whose comments made a significant contribution to improving this final version. We thank Alheit du Toit for the collection and sequencing of wonder worm specimens.

Additional Information and Declarations

Competing Interests

Author Contributions

Field Study Permissions

DNA Deposition

Data Availability

New Species Registration

The authors declare there are no competing interests.

Jyothi Kara and Isabel C. Molina-Acevedo conceived and designed the experiments, performed the experiments, analyzed the data, prepared figures and/or tables, authored or reviewed drafts of the paper, and approved the final draft.

Joana Zanol conceived and designed the experiments, analyzed the data, authored or reviewed drafts of the paper, and approved the final draft.

Carol Simon and Izwandy Idris analyzed the data, authored or reviewed drafts of the paper, and approved the final draft.

The following information was supplied relating to field study approvals (i.e., approving body and any reference numbers):

The Department of Agriculture, Forestry and Fisheries approved the collection of marine invertebrates (RES2019-49).

The following information was supplied regarding the deposition of DNA sequences:

The newly generated sequences in this study for Marphysa sherlockae n. sp. are available at GenBank: MT840349, MT840350, MT840351.

The following information was supplied regarding data availability:

The raw data are available in Data S2, S3, S4, S5.

The following information was supplied regarding the registration of a newly described species:

Publication LSID: urn:lsid:zoobank.org:pub:C4C08B70-EC42-4AE1-9F9A-FDC717142D35

Marphysa sherlockae n. sp. LSID: urn:lsid:zoobank.org:act:2D2AC893-C074-46CC-B731-F0D632C66836

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
