# Peer review of "Morphological and molecular systematic review of Marphysa Quatrefages, 1865 (Annelida: Eunicidae) species from South Africa"

_PeerJ, doi:10.7717/peerj.10076_

## Round 0.1 · original submission · Major Revisions

I have heard back from two reviewers, both of whom were positive about your work. At the same time, they have both offered numerous constructive comments that will help you make your work better. Please consider each comment carefully. As well, please ensure the English is up to international scientific standards.

Reviewer 1 ·

Basic reporting

This study reviewed the genus Marphysa of South Africa based on morphological characters and molecular information. I think that the manuscript is suitable for publication in an international journal. This is the first step for the conservation of local species in South Africa.

Some modifications are needed before accepting the manuscript.
1. The figures of pectinate chaetae and other chaetae are needed to replace. Authors use the pectinate chaetae for diagnostic characters, but the pectinate chaetae in the figures are hard to see the characters. The SEM photographs or line drawings are needed for them. Line drawings of parapodia are also helpful for understanding the diagnostic characteristics.
2. Indicating previous records in figure 1 helps to understand the distribution and taxonomic position of collected samples for readers.
3. Also, the key to species or comparative table of Marphysa in South Africa helps accurate identification for readers.
4. I recommend adding Maximum Likelihood tree.
5. In the remarks section of M. haemasoma, authors should add a paragraph for morphological comparison and explanation of taxonomic treatment between M. haemasoma and M. elityeni. Additionally, photographs of M. elityeni support comparing both species.
6. There are many typographical errors.

Best regards.

Experimental design

no comment

Validity of the findings

no comment

Additional comments

no comment

Annotated reviews are not available for download in order to protect the identity of reviewers who chose to remain anonymous.

·

Basic reporting

The paper needs to be carefully checked by the English native speakers amongst the authors

have made changes throughout and some globals are required

Experimental design

not an experimental study

Validity of the findings

good study and nice images - just need to tighten up text and also the puncutuation of the synonymies

Additional comments

see my comments on the atteched pdf

---

## Round 0.2 · Minor Revisions

I have heard back from the same two reviewers, and only minor, mainly typographical errors remain. I imagine you will be able resubmit soon, and look forward to seeing your new version.

Reviewer 1 ·

Basic reporting

Thank you for sending the modified manuscript.
I added some modifications in 'References'.

Experimental design

no comment

Validity of the findings

no comment

Additional comments

no comment

Annotated reviews are not available for download in order to protect the identity of reviewers who chose to remain anonymous.

·

Basic reporting

some minor typographical edits required

Experimental design

Ok

Validity of the findings

all Ok

Additional comments

just some minor corrections required

---

## Round 0.3 · accepted · Accept

The paper has been revised well; thank you for you attention to detail. I am pleased to accept this work and move it into production!